# Genomic epidemiology analysis of drug-resistant *Mycobacterium tuberculosis* distributed in Mexico

Paulina M. Mejía-Ponce[1], Elsy J. Ramos-González[2], Axel A. Ramos-García[1], Edgar E. Lara-Ramírez[2], Alma R. Soriano-Herrera[2], Mitzy F. Medellín-Luna[2,3], Fernando Valdez-Salazar[2], Claudia Y. Castro-Garay[2], José J. Núñez-Contreras[2], Marcos De Donato-Capote[4], Ashutosh Sharma[4], Julio E. Castañeda-Delgado[2,5], Roberto Zenteno-Cuevas[6,7], Jose Antonio Enciso-Moreno[2,8]*, Cuauhtémoc Licona-Cassani[1,7,9]*

1 Centro de Biotecnología FEMSA, Escuela de Ingeniería y Ciencias, Tecnológico de Monterrey, Nuevo León, México, 2 Unidad de Investigación Biomédica de Zacatecas, Instituto Mexicano Del Seguro Social, Zacatecas, México, 3 Posgrado en Ciencias Farmacobiológicas, Universidad Autónoma de San Luis Potosí, San Luis Potosí, México, 4 Escuela de Ingeniería y Ciencias, Tecnológico de Monterrey, Querétaro, México, 5 Consejo Nacional de Ciencia y Tecnología, CONACYT, Ciudad de México, México, 6 Instituto de Salud Pública, Universidad Veracruzana, Veracruz, México, 7 Red Multidisciplinaria de Investigación en Tuberculosis, Ciudad de México, México, 8 Facultad de Química, Universidad Autónoma de Querétaro, Querétaro, México, 9 Division of Integrative Biology, The Institute for Obesity Research, Tecnológico de Monterrey, Nuevo León, México

* joseantonio.enciso@gmail.com (JAEM); clicona@tec.mx (CLC)

**Data Availability Statement:** The raw WGS data generated in this study can be accessed in the NCBI BioProject database under the accession

## Abstract

Genomics has significantly revolutionized pathogen surveillance, particularly in epidemiological studies, the detection of drug-resistant strains, and disease control. Despite its potential, the representation of Latin American countries in the genomic catalogues of *Mycobacterium tuberculosis (Mtb)*, the bacteria responsible for Tuberculosis (TB), remains limited. In this study, we present a whole genome sequencing (WGS)-based analysis of 85 *Mtb* clinical strains from 17 Mexican states, providing insights into local adaptations and drug resistance signatures in the region. Our results reveal that the Euro-American lineage (L4) accounts for 94% of our dataset, showing 4.1.2.1 (Haarlem, n = 32), and 4.1.1.3 (X-type, n = 34) sublineages as the most prevalent. We report the presence of the 4.1.1.3 sublineage, which is endemic to Mexico, in six additional locations beyond previous reports. Phenotypic drug resistance tests showed that 34 out of 85 *Mtb* samples were resistant, exhibiting a variety of resistance profiles to the first-line antibiotics tested. We observed high levels of discrepancy between phenotype and genotype associated with drug resistance in our dataset, including pyrazinamide-monoresistant *Mtb* strains lacking canonical variants of drug resistance. Expanding the Latin American *Mtb* genome databases will enhance our understanding of TB epidemiology and potentially provide new avenues for controlling the disease in the region.

number PRJNA824124 (https://www.ncbi.nlm.nih.gov/bioproject/?term=PRJNA824124).

**Funding:** We are also grateful for the financial support provided by several grants, including the Mexican National Council for Science and Technology (CONACYT) - Ciencia de Frontera – 319590 to CLC; postgraduate scholarships to PMMP and AARG from the CONACYT. Additionally, we are thankful to Tecnologico de Monterrey for their Seed Grant to CLC, MDD and AS, which supported genome sequencing. We would also like to acknowledge StrainBiotech for their contribution of laboratory supplies, which greatly facilitated our research.

**Competing interests:** The authors declare no conflict of interest.

## Introduction

Tuberculosis (TB) is the second deadliest global pandemic (after COVID-19), having caused approximately 1.5 million fatalities in 2020 [1, 2]. A significant factor contributing to this high mortality rate is the emergence of drug-resistant *Mycobacterium tuberculosis* (*Mtb*) strains, which are responsible for 25% of annual TB-related deaths [2]. About 3–4% of the global TB cases are resistant to rifampicin or isoniazid, two of the most effective anti-TB drugs [2, 3]. The World Health Organization (WHO) has recognized the urgent need to combat this threat and has determined that new diagnostic tools must be introduced to effectively detect drug resistant *Mtb* strains by 2025. This strategy is part of the End-TB program, which aims to reduce the incidence of TB and the number of TB-related deaths by 90% and 95%, respectively, by 2035 [2].

The designation of drug resistance markers in *Mtb* strains has resulted from extensive genomic epidemiological studies performed mostly in regions where TB is highly prevalent [4, 5]. These studies have involved screening thousands of genomes, leading to the identification of more than 90 genetic variations (known as "canonical" variants) that have been experimentally confirmed to confer resistance to 13 antitubercular drugs [6, 7]. While such studies have primarily focused on countries in Europe and Asia, with only two Latin American countries (Peru and Brazil) contributing less than 15% of genomes to the databases, they are exhaustive and highly reliable. However, the applicability of these genetic associations to isolates from geographically distant regions (such as certain countries in Latin America), with different genetic backgrounds remains unclear.

Whole genome sequence (WGS) technologies have transformed the field of TB epidemiology by increasing the resolution of drug resistance genotyping and providing detailed information on sublineages and transmission groups of *Mtb* strains [8]. In several countries, WGS is now considered the gold standard method for genotyping *Mtb* clinical strains [9, 10]. However, in developing countries and regions with lower incidence of TB, traditional genotyping techniques are still used. Although this is expected, this scenario demonstrates the urgent need for increased genomic surveillance to gain a better understanding of the lineage distribution and evolutionary mechanisms driving drug resistance on a global scale.

Mexico is considered a low-burden TB country, but it ranks third among the top contributors of TB in the Americas [11]. Most of the genotyping studies of *Mtb* clinical strains in Mexico have relied on traditional techniques [12–18], resulting in fragmentary information about the phylogenomic distribution and drug-resistant phenotypes of *Mtb* isolates in the country. Although WGS-based genotypic analysis of *Mtb* strains has only recently begun in Mexico, it has already shed light on some of the genotypic characteristics of strains in two states [19, 20]. In this study, we expand upon this knowledge by presenting the genomic characterization of 85 clinical strains of *Mtb* from 17 states in Mexico. Our work provides a robust identification of sublineages and drug resistance genetic signatures. We also conducted a comprehensive examination of the disparities between phenotype and genotype of drug resistance, as well as the geographical distribution of sublineages and drug-resistant samples in the region. This study offers valuable genetic information from previously unexplored parts of the country, which can contribute to the implementation of more effective drug resistance control strategies across Mexico.

## Materials and methods

### *Mycobacterium tuberculosis* clinical strains

A total of 85 clinical samples of *Mtb* were isolated from sputum. These samples were collected between 1998 and 2021 from TB positive patients from 17 Mexican states: Nuevo Leon (n = 15), Baja California (n = 10), Jalisco (n = 10), Sinaloa (n = 7), Campeche (n = 6), Oaxaca (n = 6), Ciudad de Mexico (n = 5), Durango (n = 5), Zacatecas (n = 5), Estado de Mexico (n = 4), Nayarit (n = 3), Chiapas (n = 2), San Luis Potosi (n = 2), Yucatan (n = 2), Puebla (n = 1), Sonora (n = 1), and Veracruz (n = 1). Pre-selection criteria included relevance of drug resistance phenotype and geographical representativity across the region. Transportation and processing of samples was conducted in adherence with the biosafety recommendations established by the National Institute of Diagnosis and Epidemiological Reference (InDRE). The samples were isolated in Lowestein-Jensen medium (for two months at 37˚C) for phenotypic characterization and maintenance. Isolation and primary characterization were performed at the Biomedical Research Unit of Zacatecas (UIBMZ). The UIBMZ laboratory at Instituto Mexicano del Seguro Social (IMSS) is a national reference center for diagnosis of TB. Authors did not have access to information that could identify individual participants during or after data collection.

### Phenotypic drug sensitivity test

Phenotypic drug sensitivity test (DST) was performed with the fluorometric method BACTEC MGIT 960 (Becton-Dickinson, USA) for five first-line antibiotics: isoniazid (H), rifampicin (R), ethambutol (E), streptomycin (S) and pyrazinamide (Z). The critical concentrations used to the DST analysis were H = 0.1µg/mL, R = 1.0 µg/mL, E = 5.0 µg/mL, and S = 1.0 µg/mL. The pyrazinamide DST (100 µg/mL) was performed using the BACTEC MGIT 960 PZA kit (Becton Dickinson). Second-line antibiotics DSTs were not performed due to infrastructure limitations. The diagnostic laboratory where we performed the DST experiments is operated in conjunction with integrated Diagnostic facilitators and in coordination with the National Reference Institute for Diagnosis of Mexico, the WHO TB diagnosis program, and the Stop TB Partnership. The laboratory is certified by the InDRE and the IMSS as diagnostic reference laboratories using Standard Operating Procedures (SOP´s) with blind panels that are confirmed and periodically reported to the authority.

### Genomic DNA extraction

The genomic DNA of *Mtb* isolates was extracted as previously described [21], with minor modifications. Briefly, the biomass from two-months growth in Lowenstein-Jensen tubes (BD, BBL) was resuspended in 400µL of 1X TE buffer. Cells were inactivated by incubating the cell suspension at 80˚C for 20 min, followed by an overnight incubation at room temperature with 50µL of 10mg/mL lysozyme solution (Millipore-Sigma, USA). Cellular lysis was performed by adding 70µL of 10% SDS and 5µL of 10mg/mL Proteinase-K (Millipore-Sigma, USA), mixing and incubating at 65˚C for 10 min. Then, 100µL of 5M NaCl (Millipore-Sigma, USA) and 100µL of pre-warmed (65˚C) solution of CTAB/NaCl (40mM/0.1M) (Millipore-Sigma, USA) were added, and the suspension mixed vigorously, and incubated at 65˚C for 10 min. DNA extraction was achieved by adding an equal volume of chloroform-isoamyl alcohol (24:1) (Millipore-Sigma, USA) to the lysate, which was then mixed and centrifuged at room temperature. The aqueous phase was treated with 0.6 volumes of isopropanol (Millipore-Sigma, USA) to precipitate the DNA. The resulting DNA pellet was then washed twice with 500µL of cold 70% ethanol solution. After letting the pellet air dry for 10 min, the DNA was resuspended

in sterile DNase-free water. DNA integrity was verified by electrophoresis in a 2% agarose gel (Millipore-Sigma, USA). DNA concentration and purity were determined by spectrometry with a Nanodrop 2000 (ThermoScientific, USA) and by fluorometry with a Qubit v.3 (Invitrogen, C.A., USA).

## Whole genome sequencing and bioinformatic analysis

Genomic libraries were prepared from 1ng of high-quality genomic DNA using the Nextera DNA Flex Library Prep kit (n = 53) or the Illumina DNA Prep kit (n = 32) (Illumina C.A., USA), following the manufacturer instructions. Library quality control was checked with the DNA 1000 kit (Agilent Technologies, USA) on a Bioanalyzer 2100 (Agilent Technologies, USA). Samples that passed the quality-control were pooled and normalized according to the protocol recommendations. The pool was loaded into a 300-cycle mid-output cartridge at 2pM. Sequencing was performed at the sequencing facility (TecBASE) from Tecnologico de Monterrey, Queretaro Campus using a NextSeq 550 (lllumina, C.A., USA) in a 2×150 paired end format. Raw reads are available under NCBI BioProject **PRJNA824124**.

Quality control of the raw reads was examined using the software FastQC v0.11.9 [22] before and after applying Trimmomatic v0.33 (SLIDINGWINDOW:5:20) [23]. Reads shorter than 20bp were excluded for the subsequent analysis. Variant calling analysis was performed using the MTBseq pipeline v.1.0.3 [24] with default parameters. The MTBseq analysis provided the mapping statistics, the sublineage classification, the identification of transmission groups and a list of amended variants of the 85 *Mtb* genomes. The resulting list of variants was filtered using the following criteria: first, all variants classified as "low confidence regions" (RLC dataset) according to Marin et al. 2022 [25] were removed. Second, all those positions that, at least in one sample, contained an allele classified as Uncovered (Unc) by the MTBseq output were discarded.

## Phylogeographic analysis

A phylogenomic tree was reconstructed using a list of concatenated SNPs derived from the MTBseq pipeline, applying the parameter—w12 to define the transmission groups with a distance window of 12 SNPs. The best substitution model was determined by ModelTest-NG [26]. RaxML-NG platform [27] was used to obtain the best phylogenetic tree, with a bootstrap cutoff of 0.03. The genome of *Mycobacterium bovis* used in the phylogenomic analysis can be found under accession number ERR2659159. Edition of the phylogenetic tree was conducted with the software iTOL v6 [28]. Decimal geographic coordinates were determined for each of the Mexican states using the web portal https://www.geodatos.net. Geolocated pie chart maps were created using the Free and Open Source QGIS v.3.32 (https://www.qgis.org/es/site/index.html), using the cartographic data sourced from Natural Earth Database (Quick start kit) [29].

## Prediction of drug resistant phenotypes

Drug resistance genotype for both first- and second-line antibiotics was determined using TB-profiler v4.3.0 [4], using BCF tools as caller. Sensitivity, specificity, and positive (PPV) and negative (NPV) predictive values were determined with the "epiR" package from R. The strength of agreement between phenotype and genotype predictions of drug resistance was determined with the Cohen kappa (k) coefficient by using the "fmsb" package from R.

### Ethical concerns

We obtained informed consent (written) from all participants and/or their legal guardian(s). All the information collected from TB patients was treated confidentially. Sampling was carried out without physical intervention, with full care given to the patient's integrity. The study was approved by the National Scientific Research and Ethics Committee of the IMSS, Mexico (R-2005-3301-18, and R-2013-785-001). All the methods were performed in accordance with the guidelines and regulations established by the IMSS ethics committee.

## Results

### Epidemiology and drug resistance of *Mtb* clinical samples

In this study, we selected 85 clinical isolates of *Mtb* from sputum samples of TB-positive patients from 17 Mexican states (see **Materials and Methods** section). Overall, the mean age of the population sampled was 44.3 (range 16–85 years), with even gender distribution (male: n = 42, 49%). The most common comorbidity among the patients was Type-2 Diabetes Mellitus (T2DM) (n = 29, 34%), while only three isolates (4%) were from patients co-diagnosed with HIV. The presence of the characteristic scar of the Bacillus Calmette-Guérin (BCG) vaccine revealed that 51% (n = 43) of patients had been vaccinated against TB. Additionally, almost a third of the patients (27%, n = 23) were relapse cases of TB. **S1 Table** contains a full description of the epidemiological data of the population sampled in this study.

According to the DST, 51 samples (60%) showed susceptibility to all the antibiotics tested (first-line antibiotics). The remaining 34 samples showed a phenotype of drug resistance against H (n = 20), R (n = 14), E (n = 9), S (n = 16), and Z (n = 15), showing different profiles of resistance (**S2 Table**). Sub-classification based on clinical characteristics revealed that 14.1% of samples were multi-drug resistant TB ("MDR-TB"), including resistant profiles HR, HRS, HRE, HRSZ, and HRESZ; 14.1% were "Other" with EZ, S, and Z resistant profiles; 9.4% were isoniazid-resistant TB (HR-TB) showing H, HS, HZ, and HSZ profiles; and 2.3% were rifampicin-resistant TB (RR-TB) with profiles RESZ, and RE. Pyrazinamide DST results were inconclusive for samples I001, I002, I018, and I023.

### Lineage classification and phylogenomic analysis

We obtained high-quality sequencing reads for 85 out of 88 *Mtb* clinical isolates derived from TB-positive patients. Three samples (I005, I011 and I036) were excluded from subsequent analysis due to poor library amplification or insufficient coverage of the reference genome. The remaining 85 *Mtb* genomes had an unambiguous mean coverage of 93.8X (range: 21.59 X - 304.2 X) relative to the reference genome *Mtb* H37Rv (NC_000962.3). All sequencing-related data of the 85 *Mtb* genomes are presented in **S3 Table.**

The lineage classification showed that most of the samples (80/85, 94.1%) belonged to the Euro-American lineage (L4). Only three samples, collected from Sinaloa, Campeche, and Jalisco states, were classified as East-Asian lineage (L2) and another two samples, from Nuevo Leon and Yucatan states, were identified as Indo-Oceanic lineage (L1). Among the L4 samples, we identified various sublineages, including 4.1.1 (n = 15), 4.1.2.1 (n = 12), 4.8 (n = 11), 4.1.1.3 (n = 11), 4.3.3 (n = 10), 4.3.2 (n = 4), 4.3.4.2 (n = 3), 4.3.4.1 (n = 3), 4.4.1.1 (n = 3), 4.1.2 (n = 3), 4.7 (n = 2), 4.1.4 (n = 2), 4.3 (n = 1). In **S4 Table** we show the results of the sublineage classification for the 85 *Mtb* samples using MTBseq and TB-profiler pipelines as well as their corresponding family typing name.

We reconstructed a phylogenomic tree for the 85 *Mtb* samples using 11,150 unambiguous SNPs, excluding repetitive regions and drug resistance markers. The topology of the phylogeny

showed a concordant sample distribution according to sublineage classification (Fig 1A). We observed that 4.1.1.3 sublineage was placed within the 4.1.1 clade (X-type) with the sample I050 (from the state of Jalisco). The sublineage 4.1.1.3 is distributed across seven states (Zacatecas, Sinaloa, San Luis Potosi, Jalisco, Nuevo Leon, Estado de Mexico, and Chiapas), representing six novel locations than previously reported in the region. Withing this clade, we found one small transmission group of two identical samples (I103 and I125, genetic distance of zero SNPs) from the north of Mexico (Zacatecas and Sinaloa states, respectively). Additionally, a two-sample (I041 and I106, genetic distance of 11 SNPs) transmission group from 4.4.1.1 sublineage was found in the Pacific coastline including Baja California and Nayarit states.

The geographical distribution of the 85 *Mtb* samples exhibiting their sublineage classification is shown in Fig 1B. The largest number of samples (43/85) was obtained from the north region of the country, which included three main sublineages: 4.8 (n = 9), 4.1.1 (n = 9), and 4.1.1.3 (n = 7). The central region also showed prevalence of 4.1.1 (n = 6) and 4.1.1.3 (n = 2)

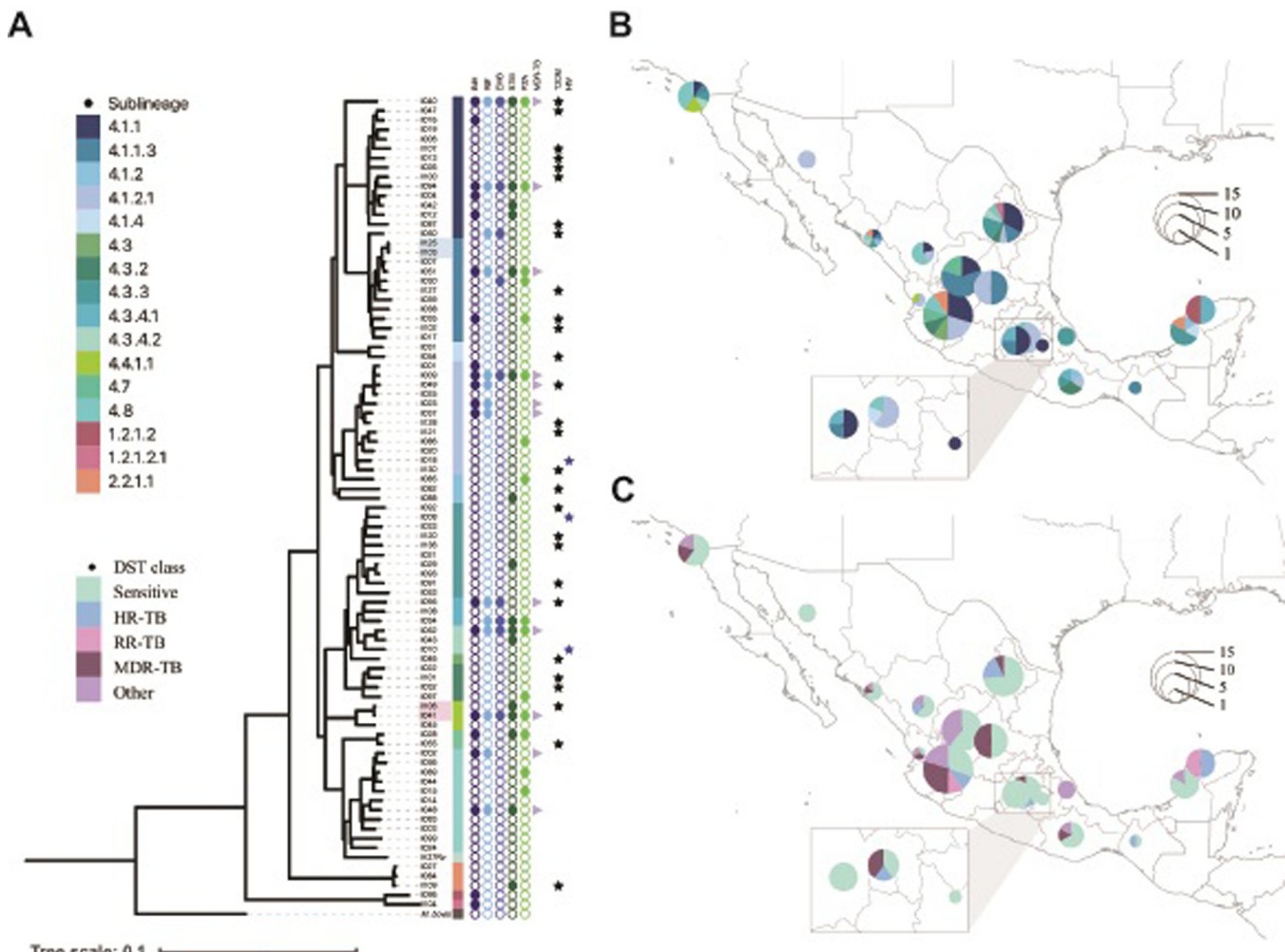

**Fig 1. Phylogenomic tree of the 85 *Mtb* clinical isolates from Mexico.** (A) The phylogeny was reconstructed with 11,150 SNPs, the GTR+FO+G4m substitution model, and a bootstrap cutoff of 0.03. Color strip indicates the classification by sublineage, followed by the phenotypic DST and the comorbidities T2DM (black star) and HIV (blue star) associated with each sample. Colored clades (light blue and pink) indicate transmission groups. (B) Geographical distribution of *Mtb* samples, colored by sublineage and (C) drug-resistant classification.

sublineages, but most samples belonged to the 4.1.2.1 (n = 8) sublineage. In the southern region, the 4.3.3 sublineage (n = 4) was the most abundant in our dataset.

Among the MDR-TB samples in our dataset, the 4.1.2.1 sublineage (n = 4) was the most representative. However, we found no significant correlation between any drug-resistance profile and sublineages, including the HRESZ-resistance profile (Spearman correlation, *r* = 0, *p-value* = 0.559). The geographical distribution of clinical drug-resistant classifications, based on phenotypic DST of the samples, is presented in **Fig 1C**. We observed that at least one sample per state (excluding sites with only one sample) exhibited drug resistance to at least one of the first-line antibiotics tested. This underscores the importance of surveillance for drug-resistant TB infections in the country.

## Genotypic analysis of drug-resistance phenotypes

For readability, we use the term "canonical variants" to refer to all variants associated with drug resistance (both first- and second-line antibiotics) identified by the automated platform described in the **Materials and Methods** section. We identified 47 canonical variants in 25 out of 85 *Mtb* isolates (**Table 1**). Eight samples (I018, I023, I030, I037, I049, I051, I096, and I099) carried double canonical variants associated with H-, R- and fluoroquinolones (FQs) resistance, and one of them (I096) harbored a triple mutation related with H-resistance.

In our dataset, the most frequent canonical variants associated with H resistance were *ahpC* Asp73His (n = 5), *katG* Ser315Thr (n = 4), and *fabG1* –15C>T (n = 4), while *rpoB* His445Asn (n = 3), and *rpoB* Leu452Pro (n = 3) were the most prevalent mutations related to R resistance. We also identified several mutations associated with second-line antibiotics, including single (n = 3), and double (n = 1) mutations related with FQs, one mutation associated with aminoglycosides (AMGs), five associated with ethionamide (ETO) and another one related with delamanid (DMD) resistance. It is worth noting that we identified ten out of 47 canonical variants classified as "Uncertain" (group 3) in the WHO's catalogue of drug-resistance-associated mutations. Among these uncertain mutations, we found *ahpC* Asp73His present in five samples with an H-susceptible phenotype. The remaining uncertain mutations, except *gid* Leu79Ser and *katG* Ser315Gly, appeared with another canonical variant belonging to "Associated with resistance" (group 1) or "Associated with resistance interim" (group 2) in phenotypically resistant samples. In addition, *katG* Gly428Arg, *katG* 9–16 GCAACACC deletion, and *ethA* Glu223Lys variants were not listed within the WHO's catalogue. Both *katG* mutations co-occurred with other canonical variants classified as uncertain in the same sample, but just deletion 9–16 GCAACACC (plus *ahpC* -54C>T and *katG* Ser140Asn) was present in a H-resistant sample.

## Correlation analysis between phenotypic and genotypic drug resistance

We explored the level of concordance between genotypes and DST in our dataset. Our findings showed low sensitivity (<55%) and PPV (<67%) for all tested antibiotics, while specificity and NPV were higher than 90% and 80%, respectively (**Table 2**). Regarding the strength of agreement (k value), we observed a *Moderate* agreement for H in our dataset. However, R exhibited a *Fair* agreement, and E, S, and Z showed *No agreement* to *Slight* agreement. For MDR-TB samples, we obtained a *Moderate* agreement (44%) with 50% sensitivity and 93% specificity. We observed that removing uncertain variants improved some concordance values for H, E, and MDR-TB predictions, but not for streptomycin. **S5 Table** includes the concordance analysis using the complete dataset of canonical variants and without uncertain variants.

Next, we investigated the two types of phenotypic disagreements present in our samples: i) samples that were phenotypically resistant but lacked canonical variants or encoded uncertain

**Table 1. Canonical variants associated with drug resistance of the 85 *Mtb* isolates.**

| Drug | Canonical variants | WHO-classification | Total frequency | % Resistant samples | % Susceptible samples | Previous report in the region |
|---|---|---|---|---|---|---|
| H | *ahpC* Asp73His | Unc | 5 | 0% | 100% | [20] |
| | *fabG1* -15C>T | AwR | 4 | 75% | 25% | |
| | *inhA* -154G>A | AwR-i | 2 | 50% | 50% | |
| | *inhA* Ser94Ala | AwR-i | 1 | 0% | 100% | |
| | *katG* Ser315Gly | Unc | 1 | 100% | 0% | |
| | *katG* Ser315Thr | AwR | 4 | 50% | 50% | [19, 20] |
| | *ahpC* -52C>T | Unc | 1 | 100% | 0% | |
| | *fabG1* -8T>C | AwR-i | | | | |
| | *ahpC* -52C>T | Unc | 1 | 100% | 0% | |
| | *katG* Gly428Arg | NA | | | | |
| | *ahpC* Asp73His | Combo | 2 | 50% | 50% | |
| | *katG* Ser315Thr | | | | | |
| | *inhA* -154G>A | AwR | 1 | 100% | 0% | |
| | *inhA* Ile21Val | Unc | | | | |
| | *katG* Ala379Val | Unc | 1 | 100% | 0% | |
| | *katG* Ser315Asn | AwR | | | | |
| | *ahpC* -54C>T | Unc | 1 | 0% | 100% | |
| | *katG* 9_16delGCAACACC | NA | | | | |
| | *katG* Ser140Asn | Unc | | | | |
| R | *rpoB* Asp435Val | AwR | 2 | 100% | 0% | [19, 20] |
| | *rpoB* His445Asn | AwR | 3 | 33% | 67% | |
| | *rpoB* His445Asp | AwR | 1 | 0% | 100% | [19] |
| | *rpoB* His445Cys | AwR | 1 | 0% | 100% | |
| | *rpoB* Leu452Pro | AwR | 3 | 67% | 33% | [20] |
| | *rpoB* Ser441Leu | AwR | 1 | 100% | 0% | |
| | *rpoB* Ser450Leu | AwR | 1 | 0% | 100% | [19] |
| | *rpoB* His445Arg | AwR | 1 | 0% | 100% | |
| | *rpoB* Ser450Leu | AwR | | | | |
| E | *embA* -16C>T | Unc | 1 | 0% | 100% | |
| | *embB* Gly406Ala | AwR | 1 | 0% | 100% | |
| | *embB* Met306Ile | AwR | 3 | 33% | 67% | [19] |
| | *embB* Met306Leu | AwR | 1 | 0% | 100% | |
| | *embB* Met306Val | AwR | 2 | 50% | 50% | [19, 20] |
| S | *gid* Gln127* | AwR-i | 1 | 0% | 100% | |
| | *gid* Leu79Ser | Unc | 1 | 100% | 0% | |
| | *rpsL* Lys43Arg | AwR | 1 | 0% | 100% | [19] |
| | *rpsL* Lys88Arg | AwR | 1 | 0% | 100% | |
| | *rrs* 514A>C | AwR | 1 | 0% | 100% | |
| Z | *pncA* 118dupG | AwR-i | 1 | NA | NA | |
| | *pncA* 390_391dupGG | AwR-i | 1 | 100% | 0% | |
| | *pncA* Gly132Ser | AwR | 1 | NA | NA | |
| | *pncA* His71Arg | AwR | 1 | 0% | 100% | |
| | *pncA* Thr61Pro | Unc | 1 | NA | NA | |
| | *pncA* Trp68* | AwR | 1 | 100% | 0% | [19] |

(*Continued*)

**Table 1.** (Continued)

| Drug | Canonical variants | WHO-classification | Total frequency | % Resistant samples | % Susceptible samples | Previous report in the region |
|---|---|---|---|---|---|---|
| FQs | *gyrA* Asp94Ala | AwR | 1 | NA | NA | |
| | *gyrA* Asp94Gly | AwR | 1 | NA | NA | [20] |
| | *gyrA* Gly88Ala | AwR-i | 1 | NA | NA | |
| | *gyrA* Asp94His | AwR | 1 | NA | NA | |
| | *gyrA* Ser91Pro | AwR | | | | |
| AMGs | *rrs* 1484G>T | AwR | 1 | NA | NA | |
| ETO | *ethA* Glu223Lys | NA | 4 | NA | NA | |
| | *fabG1* -15C>T | AwR | 4 | NA | NA | |
| | *fabG1* -8T>C | AwR-i | 1 | NA | NA | |
| | *inhA* -154G>A | AwR-i | 3 | NA | NA | |
| | *inhA* Ser94Ala | AwR-i | 1 | NA | NA | |
| DMD | *ddn* Leu49Pro | AwR-i | 3 | NA | NA | |

*Stop codon; AwR = Associated with Resistance (group 1); AwR-i = Associated with Resistance interim (group 2); Unc = Uncertain (group3); Combo = Combinatorial (group 6); NA = Not Available.

variants ($R_S$), and ii) samples that were phenotypically susceptible with canonical variants ($S_R$). Our analysis revealed that 45% of H-resistant samples and 57% of R-resistant samples exhibited $R_S$ discordance. Interestingly, for E, S, and Z, the proportion of $R_S$ discordance was even higher (reaching 78%, 100%, and 87%, respectively). Additionally, we identified three H-susceptible samples (I030, I099, and I106) carrying one of the most prevalent canonical variants associated with H resistance (*katG* Ser315Thr). Furthermore, we detected seven mutations outside the 81-bp Rifampicin-Resistance Determining Region (RRDR) of the *rpoB* gene in an equal number of R-susceptible samples. A comprehensive list of the samples exhibiting both types of discordance, as well the specific canonical variants involved, is shown in **S6 Table**.

## Discussion

In 2019, Mexico reported an estimated of 29,000 cases of TB, with 3.3% of them classified as drug-resistant infections [30]. While previous studies have investigated the geographic distribution of TB cases in the country [13, 31–35], there is little information prevalent lineages and drug resistance patterns. Given the detection of latent TB in migrants traveling through different routes in Mexico, understanding the distribution of TB cases is critical [36]. In this study, we present a comprehensive genomic characterization of 85 clinical samples of *Mtb* from various locations in Mexico using WGS. We provide robust sublineage classification and identify small transmission groups in the region. Furthermore, we describe the canonical variants

**Table 2. Concordance analysis of drug resistance in the 85 *Mtb* samples.**

| Drug | Sensitivity | Specificity | PPV | NPV | k value, interpretation |
|---|---|---|---|---|---|
| H | 0.55 (0.32, 0.77) | 0.89 (0.79, 0.96) | 0.61 (0.36, 0.83) | 0.87 (0.76, 0.94) | 0.46 (0.22, 0.70), Moderate |
| R | 0.43 (0.18, 0.71) | 0.90 (0.81, 0.96) | 0.46 (0.19, 0.75) | 0.89 (0.79, 0.95) | 0.34 (0.04, 0.64)), Fair |
| E | 0.22 (0.03, 0.60) | 0.93 (0.85, 0.98) | 0.29 (0.04, 0.71) | 0.91 (0.82, 0.96) | 0.17 (-0.26, 0.61), Slight |
| S | 0.00 (0.00, 0.21) | 0.94 (0.86, 0.98) | 0.00 (0.00, 0.60) | 0.80 (0.70, 0.88) | -0.08(-0.50, 0.33), No agreement |
| Z | 0.13 (0.02, 0.40) | 0.98 (0.92, 1.00) | 0.67 (0.09, 0.99) | 0.83 (0.73, 0.91) | 0.17 (-0.22, 0.57), Slight |
| MDR-TB | 0.5 (0.21, 0.79) | 0.93 (0.85, 0.98) | 0.55 (0.23, 0.83) | 0.92 (0.83, 0.97) | 0.44 (0.14, 0.75), Moderate |

associated with drug resistance present in our dataset and analyze the most relevant discrepancies related to their phenotype.

One of the most prevalent lineages of *Mtb* in our dataset is L4, which has a global distribution, including several countries in Latin America [37]. It is presumed that L4 was introduced to this region from Europe during the colonization of The Americas, and local adaptation played a crucial role in its successful prevalence [38]. Regarding to L4 sublineages, we identified 11 isolates classified as 4.1.1.3 (X-type), a sublineage previously reported as endemic in Mexico with strong associations to drug resistance [19, 20, 39]. Although drug resistance was detected in only three of these samples (profiles: EZ, HZ, and HRSZ), we identified a small transmission group within this sublineage. Localized analysis of samples from Veracruz state had previously reported transmission groups within the 4.1.1.3 sublineage [39]. These isolates were identified in most of the sites sampled which strongly suggest that the 4.1.1.3 sublineage may be widely distributed in Mexico and warrants close monitoring in the region due to its high transmissibility compared to other sublineages.

Regarding drug-resistant *Mtb* isolates, we found that 14.1% of the samples were classified as MDR-TB, which is a proportion close to the mean (15.8%) found in a previous review conducted in Mexico in 2014 [40]. However, more recent studies have reported varying levels of MDR-TB in Mexico, ranging from 6% (2/32) to 57% (46/81) in Jalisco and Veracruz states, respectively [19, 20]. Furthermore, it is worthy to mention that most of the drug resistance phenotypic tests conducted in Mexico only cover first-line antibiotics [13, 14, 18, 31, 41]. Only one study has reported second-line antibiotics DST assays in Mexico [42]. While this information may be biased due to the small number of samples tested, it suggests that efforts to diagnose MDR-TB and XDR-TB in the region need to be increased.

Resistance to the antibiotic Z is commonly observed in MDR-TB isolates [43]. In our dataset, we found that 50% of MDR-TB samples were resistant to Z, and five out of 85 *Mtb* isolates exhibited Z-monoresistance. Notably, none of these five samples had canonical variants within the *pncA* gene. These results correlate with previous studies in Mexico, which have also reported Z-monoresistant *Mtb* isolates without *pncA* variants [19, 20]. While some mycobacteria, such as *M. bovis*, are intrinsically resistant to Z [44], several *Mtb* isolates exhibiting Z-monoresistance have been reported in the United States and Canada with specific mutations in the *pncA* gene [45, 46]. A recent study found that the *clpC1* Val63Ala variant is associated with low levels of Z-monoresistance in Indo-Oceanic lineage (L1) isolates [47]. However, none of the Z-monoresistant samples in the dataset here reported presented mutations in the *clpC1* gene, and all of them belonged to the L4 lineage (including 4.8, 4.1.2, 4.3.2, and 4.1.2.1 sublineages). We discovered that among the eight Z-monoresistant samples previously reported in Veracruz and Jalisco [19, 20], two belong to the L1 lineage (specifically, the 1.2.1.2.1, and 1.2.1 sublineages), while the remaining samples were classified as belonging to the L4 lineage. These findings suggest that Z-monoresistance in Mexico may be driven by variants beyond *pncA* and *clpC1* genes, highlighting the need for more genomic-based studies in the region.

In addition, we discovered significant disparities between the DST and the WGS-based genotypic predictions of drug resistance. We analyzed *Mtb* samples with high-quality DST results according to the national and international standards (InDRE and WHO programs, respectively). To ensure the accuracy of our findings, we validated the canonical variants from TB-profiler against the latest version of the WHO catalogue of resistance-associated variants [6]. During this comparison, we identified five canonical variants classified as "uncertain", including the *ahpC* Asp73His variant, which was the most frequent in susceptible strains. Although mutations in the regulatory region of *ahpC* gene have been reported as compensatory mutations that alleviate the oxidative stress caused by the *katG* mutation [48–51], the specific *ahpC* Asp73His mutation has not been associated with drug resistance [52]. Therefore, we

did not include this variant, or any of the other uncertain variants, in our analysis of discrepancies. Despite these adjustments, our sensitivity values for H- and R-resistance were lower than those reported in previous studies in the region [19, 20].

Additionally, our Mtb dataset includes samples that show disparities in both SR and RS types, as detailed in S6 Table. Within this set, several AwR mutations were identified in H- and R-susceptible strains. For example, two samples in our dataset displayed an H-susceptible phenotype but encoded one of the most prevalent and statistically significant variants associated with H-resistance: katG Ser315Thr [6]. This particular mutation has been frequently reported globally in H-resistant samples [53, 54]. However, it's important to note that a 2002 study initially identified three Mtb samples as phenotypically H-susceptible, despite encoding the katG Ser315Thr mutation [55]. When the DST experiment was repeated, these samples were reclassified as H-resistant. This highlights the intricacies involved in interpreting susceptibility profiles and emphasizes the importance of rigorous validation using larger sample sets. The limitations of DST methods, such as the use of single concentration tubes and the selection of fittest strains during DNA extraction, may account for the observed discrepancies between phenotype and genotype. Additionally, heteroresistance, the presence of susceptible and resistant strains in one sample [56], can also affect the results of phenotypic DST [57]. We acknowledge the limited sample size in our study and advocate for the inclusion of genomes from underrepresented geographic regions. We also recommend implementing technical controls to enhance the accuracy of antibiotic resistance detection and the use of random sampling methods. This will allow for a more confident assertion that Mexico has a higher discrepancy between genotyping and phenotyping DST compared to other regions Furthermore, our study contributes to a better understanding of the sublineages circulating in Mexico and their potential for their inclusion in global databases and comprehensive studies of drug resistance.

## Supporting information

**S1 Table. Epidemiological data from the 85 *Mtb* genomes.**
(XLSX)

**S2 Table. Drug susceptibility test results of the 85 *Mtb* genomes.**
(XLSX)

**S3 Table. Whole genome sequencing information from the 85 *Mtb* genomes.**
(XLSX)

**S4 Table. Sublineage classification of the 85 *Mtb* samples by MTBseq and TB-profiler tools.**
(XLSX)

**S5 Table. Concordance analysis of the phenotypic DST and canonical variants of the 85 *Mtb* samples.**
(XLSX)

**S6 Table. Discrepancies between the phenotype and genotype prediction of drug resistance from the 85 *Mtb* samples.**
(XLSX)

## Acknowledgments

We would like to express our deepest gratitude to the individuals who generously participated in this study, as well as to the clinical personnel at all sample collection sites for their invaluable

assistance. Additionally, we are thankful to the Centro de Biotecnologia FEMSA from Tecnologico de Monterrey and StrainBiotech to provide support to perform the research.

## Author Contributions

**Conceptualization:** Edgar E. Lara-Ramírez, Julio E. Castañeda-Delgado, Jose Antonio Enciso-Moreno, Cuauhtémoc Licona-Cassani.

**Data curation:** Paulina M. Mejía-Ponce, Cuauhtémoc Licona-Cassani.

**Formal analysis:** Paulina M. Mejía-Ponce, Julio E. Castañeda-Delgado, Jose Antonio Enciso-Moreno, Cuauhtémoc Licona-Cassani.

**Funding acquisition:** Marcos De Donato-Capote, Ashutosh Sharma, Julio E. Castañeda-Delgado, Jose Antonio Enciso-Moreno, Cuauhtémoc Licona-Cassani.

**Investigation:** Paulina M. Mejía-Ponce, Elsy J. Ramos-González, Axel A. Ramos-García, Edgar E. Lara-Ramírez, Alma R. Soriano-Herrera, Mitzy F. Medellín-Luna, Fernando Valdez-Salazar, Claudia Y. Castro-Garay, José J. Núñez-Contreras, Julio E. Castañeda-Delgado, Roberto Zenteno-Cuevas, Cuauhtémoc Licona-Cassani.

**Methodology:** Paulina M. Mejía-Ponce, Axel A. Ramos-García, Alma R. Soriano-Herrera, Mitzy F. Medellín-Luna, Fernando Valdez-Salazar, Claudia Y. Castro-Garay, José J. Núñez-Contreras, Julio E. Castañeda-Delgado, Roberto Zenteno-Cuevas, Jose Antonio Enciso-Moreno, Cuauhtémoc Licona-Cassani.

**Project administration:** Cuauhtémoc Licona-Cassani.

**Supervision:** Julio E. Castañeda-Delgado, Jose Antonio Enciso-Moreno, Cuauhtémoc Licona-Cassani.

**Writing – original draft:** Paulina M. Mejía-Ponce, Cuauhtémoc Licona-Cassani.

**Writing – review & editing:** Paulina M. Mejía-Ponce, Julio E. Castañeda-Delgado, Roberto Zenteno-Cuevas, Jose Antonio Enciso-Moreno, Cuauhtémoc Licona-Cassani.

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
