## [Decision Letter · Decision Letter 0]

30 Aug 2023

PONE-D-23-08010Genomic epidemiology analysis of drug-resistant Mycobacterium tuberculosis distributed in MexicoPLOS ONE

Dear Dr. Licona Cassani,

Thank you for submitting your manuscript to PLOS ONE. After careful consideration, we feel that it has merit but does not fully meet PLOS ONE’s publication criteria as it currently stands. Therefore, we invite you to submit a revised version of the manuscript that addresses the points raised during the review process.

We look forward to receiving your revised manuscript.

Kind regards,

Dwij Raj Bhatta, PhD

Academic Editor

PLOS ONE

Journal Requirements:

2. Thank you for providing the ethics approval docuemnts. It seems the approval numbers in the documents do not fully match the ones in your ethics statement (R-2005-3301-18, and R-2013-785-001). Please comment on this and clarify if there are any missing ethics documents.

In addition, please provide copies of their English translation and clarify if clinical samples of Mtb were collected by any of the authors.

Please also indicate if the provided ethics approval documents have a expiration date and clarify why data access (in Novermber 2022) is a few years after the ethics approval (in 2018).

"We are also grateful for the financial support provided by several grants, including the Mexican National Council for Science and Technology (CONACYT) - Ciencia de Frontera – 319590 to CLC; postgraduate scholarships to PMMP and AARG from the CONACYT. Additionally, we are thankful to Tecnologico de Monterrey for their Seed Grant to CLC, MDD and AS, which supported genome sequencing. We would also like to acknowledge StrainBiotech for their contribution of laboratory supplies, which greatly facilitated our research."

"..We are also grateful for the financial support provided by several grants, including the Mexican National Council for Science and Technology (CONACYT) - Ciencia de Frontera – 319590 to CLC; postgraduate scholarships to PMMP and AARG from the CONACYT. Additionally, we are thankful to Tecnologico de Monterrey for their Seed Grant to CLC, MDD and AS, which supported genome sequencing.."

"We are also grateful for the financial support provided by several grants, including the Mexican National Council for Science and Technology (CONACYT) - Ciencia de Frontera – 319590 to CLC; postgraduate scholarships to PMMP and AARG from the CONACYT. Additionally, we are thankful to Tecnologico de Monterrey for their Seed Grant to CLC, MDD and AS, which supported genome sequencing. We would also like to acknowledge StrainBiotech for their contribution of laboratory supplies, which greatly facilitated our research."

7. We note that you have stated that you will provide repository information for your data at acceptance. Should your manuscript be accepted for publication, we will hold it until you provide the relevant accession numbers or DOIs necessary to access your data. If you wish to make changes to your Data Availability statement, please describe these changes in your cover letter and we will update your Data Availability statement to reflect the information you provide.

8. Please amend either the abstract on the online submission form (via Edit Submission) or the abstract in the manuscript so that they are identical.

9. We note that Figure 1 in your submission contain map images which may be copyrighted. All PLOS content is published under the Creative Commons Attribution License (CC BY 4.0), which means that the manuscript, images, and Supporting Information files will be freely available online, and any third party is permitted to access, download, copy, distribute, and use these materials in any way, even commercially, with proper attribution. For these reasons, we cannot publish previously copyrighted maps or satellite images created using proprietary data, such as Google software (Google Maps, Street View, and Earth). For more information, see our copyright guidelines: http://journals.plos.org/plosone/s/licenses-and-copyright.

(1) You may seek permission from the original copyright holder of Figure 1 to publish the content specifically under the CC BY 4.0 license.  

**Additional Editor Comments:**

Very important country specific scientific data / findings on TB drug resistance highlighted in submitted paper & has a global importance ! However, athours need to adress queries of reviwers before publication

Reviewers' comments:

Reviewer's Responses to Questions

**Comments to the Author**

1. Is the manuscript technically sound, and do the data support the conclusions?

Reviewer #1: Partly

Reviewer #2: Yes

2. Has the statistical analysis been performed appropriately and rigorously? 

Reviewer #1: N/A

Reviewer #2: N/A

3. Have the authors made all data underlying the findings in their manuscript fully available?

Reviewer #1: Yes

Reviewer #2: Yes

4. Is the manuscript presented in an intelligible fashion and written in standard English?

Reviewer #1: Yes

Reviewer #2: Yes

5. Review Comments to the Author

Reviewer #1: The author can detail on the sample collection - If all positive cultures from 17 states between 1998-2021 was analyzed in this study or it had any other preselection criteria.

2.The authors have done the phenotypic dst for 1st line drugs alone to look into the pDST and gDST correlation with only few strains for looking into the correlation. This interpretation may change with more no.of strains being analysed

3. The no of AwR mutations from the catalogue that were susceptible to the respective 1st line drugs in this study can be looked into in terms of these mutations that were catagorized as AwR based on Expert rules in the WHO Catalogue and discussed.

4. Based on the findings from this study, the need for analysing country specific data set and analysis framework for the mutations association with drug resistance can be recommended to help with a meaningful recommendation at the global level.

Reviewer #2: Congratulation Paulina M et al for the nice study in the field of drug resistant tuberculosis.

The study on the genomic epidemiology of drug-resistant Mycobacterium tuberculosis distributed in Mexico is an important contribution to the field of tuberculosis research. The authors have undertaken a comprehensive analysis using genomic data to shed light on the prevalence, distribution, and potential transmission patterns of drug-resistant strains of M. tuberculosis in the Mexico. Authors have also performed whole- Genome Sequencing of 85 isolates of M. tuberculosis isolated from different parts of Mexico. The results of this study could have significant implications for tuberculosis control strategies and public health interventions in Mexico.

The methodology appears to be sound and well-validated, I am fully agree on this .

The findings of the research has the potential to contribute valuable information to global efforts in combating tuberculosis and drug resistance. Sharing insights on the genomic diversity and transmission patterns of M. tuberculosis strains in Mexico can aid in a better understanding of the global dissemination of drug resistance.

Although the study focused on genomic analysis, it would be beneficial to include experimental data to validate drug resistance mechanisms identified through genomics. This would strengthen the overall conclusions and interpretations of the study. Furthermore, geospatial data is lacking. Addition of this data provides regional variations in drug-resistant tuberculosis distribution and identify specific hotspots of DR TB, which will in fact guide targeted interventions.

Please include details clinical information (e.g., treatment outcomes, patient demographics) with genomic data can provide a more comprehensive view of the epidemiological dynamics of drug-resistant tuberculosis in Mexico. This would offer valuable insights into risk factors and patient-specific management strategies.

It is suggested for sharing the raw genomic data through public repositories .

The study on genomic epidemiology analysis of drug-resistant Mycobacterium tuberculosis distributed in Mexico represents a significant contribution to the field of tuberculosis research. The authors have conducted a thorough investigation, and their findings have important implications for public health. Addressing the suggestions for improvement would strengthen the study's overall impact and contribute to its broader relevance and applicability in combating drug-resistant tuberculosis both in Mexico and globally. Therefore, I would like to recommend this study for publication after reviewing the reviewer suggestion.

Thank you

Regenerate

6. PLOS authors have the option to publish the peer review history of their article (what does this mean?). If published, this will include your full peer review and any attached files.

Reviewer #1: No

Reviewer #2: **Yes: **Komal Raj Rijal

---

## [Author Response · Author response to Decision Letter 0]

18 Sep 2023

Journal Requirements:

and

We have reviewed PLOS ONE style requirements and have modified the manuscript accordingly.

2. Thank you for providing the ethics approval docuemnts. It seems the approval numbers in the documents do not fully match the ones in your ethics statement (R-2005-3301-18, and R-2013-785-001). Please comment on this and clarify if there are any missing ethics documents.

In addition, please provide copies of their English translation and clarify if clinical samples of Mtb were collected by any of the authors.

Please also indicate if the provided ethics approval documents have a expiration date and clarify why data access (in Novermber 2022) is a few years after the ethics approval (in 2018).

Thanks for the observation. We modified the ethics statement (See below) and now all the missing ethics documents are provided (in English). 

The data access was made possible until 2022 due to financial problems encountered in the laboratory, including laboratory restrictions due to COVID19 pandemic.

“Ethical concerns

We obtained informed consent (written) from all participants and/or their legal guardian(s). All the information collected from TB patients was treated confidentially. Sampling was carried out without physical intervention, with full care given to the patient's integrity. None of the authors were involved in the sampling procedure. The study was approved by the National Scientific Research and Ethics Committee of the IMSS, Mexico (R-2005-3301-18, R-2013-785-001 and R-2018-785-118). The ruling for each ethics protocol is valid for one year upon specific request of an extension. All the methods were performed in accordance with the guidelines and regulations established by the IMSS ethics committee.”

Thanks for the observation. Changes have been made accordingly.

"We are also grateful for the financial support provided by several grants, including the Mexican National Council for Science and Technology (CONACYT) - Ciencia de Frontera – 319590 to CLC; postgraduate scholarships to PMMP and AARG from the CONACYT. Additionally, we are thankful to Tecnologico de Monterrey for their Seed Grant to CLC, MDD and AS, which supported genome sequencing. We would also like to acknowledge StrainBiotech for their contribution of laboratory supplies, which greatly facilitated our research."

Thanks for the observation. We have modified the “financial information” in the submission platform. 

In addition, we would like to mention that, as stated in the “author contribution” section, all funder authors participated actively in the work.

"..We are also grateful for the financial support provided by several grants, including the Mexican National Council for Science and Technology (CONACYT) - Ciencia de Frontera – 319590 to CLC; postgraduate scholarships to PMMP and AARG from the CONACYT. Additionally, we are thankful to Tecnologico de Monterrey for their Seed Grant to CLC, MDD and AS, which supported genome sequencing.."

"We are also grateful for the financial support provided by several grants, including the Mexican National Council for Science and Technology (CONACYT) - Ciencia de Frontera – 319590 to CLC; postgraduate scholarships to PMMP and AARG from the CONACYT. Additionally, we are thankful to Tecnologico de Monterrey for their Seed Grant to CLC, MDD and AS, which supported genome sequencing. We would also like to acknowledge StrainBiotech for their contribution of laboratory supplies, which greatly facilitated our research."

Thanks for the observation. We have modified the acknowledgment section as follow:

“Acknowledgements

We would like to express our deepest gratitude to the individuals who generously participated in this study, as well as to the clinical personnel at all sample collection sites for their invaluable assistance. Additionally, we are thankful to the Centro de Biotecnologia FEMSA from Tecnologico de Monterrey and StrainBiotech to provide support to perform the research.”

We have modified the submission, so all the data is available. We additionally modified the Data Availability statement in the manuscript as follow:

“The raw data from this study are available in the NCBI database, Bioproject PRJNA824124 (https://www.ncbi.nlm.nih.gov/bioproject/?term=PRJNA824124).”

7. We note that you have stated that you will provide repository information for your data at acceptance. Should your manuscript be accepted for publication, we will hold it until you provide the relevant accession numbers or DOIs necessary to access your data. If you wish to make changes to your Data Availability statement, please describe these changes in your cover letter and we will update your Data Availability statement to reflect the information you provide.

We have modified the submission, so all the data is available. We additionally modified the Data Availability statement in the manuscript as follow:

“The raw data from this study are available in the NCBI database, Bioproject PRJNA824124 (https://www.ncbi.nlm.nih.gov/bioproject/?term=PRJNA824124).”

8. Please amend either the abstract on the online submission form (via Edit Submission) or the abstract in the manuscript so that they are identical.

Done.

9. We note that Figure 1 in your submission contain map images which may be copyrighted. All PLOS content is published under the Creative Commons Attribution License (CC BY 4.0), which means that the manuscript, images, and Supporting Information files will be freely available online, and any third party is permitted to access, download, copy, distribute, and use these materials in any way, even commercially, with proper attribution. For these reasons, we cannot publish previously copyrighted maps or satellite images created using proprietary data, such as Google software (Google Maps, Street View, and Earth). For more information, see our copyright guidelines: http://journals.plos.org/plosone/s/licenses-and-copyright.

We have modified a new version of the Figure 1 that has no restrictions of copyright.

Reviewer #1:

The author can detail on the sample collection - If all positive cultures from 17 states between 1998-2021 was analyzed in this study or it had any other preselection criteria.

The strain collection obtained includes more than 500 isolates. Samples reported in this study were pre-selected for relevant drug resistance phenotypes. We have included such information in the corresponding methodology section.

2.The authors have done the phenotypic dst for 1st line drugs alone to look into the pDST and gDST correlation with only few strains for looking into the correlation. This interpretation may change with more no.of strains being analysed.

3. The no of AwR mutations from the catalogue that were susceptible to the respective 1st line drugs in this study can be looked into in terms of these mutations that were catagorized as AwR based on Expert rules in the WHO Catalogue and discussed.

4. Based on the findings from this study, the need for analysing country specific data set and analysis framework for the mutations association with drug resistance can be recommended to help with a meaningful recommendation at the global level.

Thanks for the reviewer comment. We have included a couple of sentences on this regard in the discussion section. 

“Despite the limited number of samples analyzed in our study, our study highlights the necessity to examine data specific to individual countries and to adopt an analysis framework for the correlation between mutations and drug resistance. Such studies are essential for offering impactful recommendations on an international scale. Furthermore, our study contributes to a better understanding of the sublineages circulating in Mexico and their potential for their inclusion in global databases and comprehensive studies of drug resistance.”

Reviewer #2:

Congratulation Paulina M et al for the nice study in the field of drug resistant tuberculosis. The study on the genomic epidemiology of drug-resistant Mycobacterium tuberculosis distributed in Mexico is an important contribution to the field of tuberculosis research. The authors have undertaken a comprehensive analysis using genomic data to shed light on the prevalence, distribution, and potential transmission patterns of drug-resistant strains of M. tuberculosis in the Mexico. Authors have also performed whole- Genome Sequencing of 85 isolates of M. tuberculosis isolated from different parts of Mexico. The results of this study could have significant implications for tuberculosis control strategies and public health interventions in Mexico.

The methodology appears to be sound and well-validated, I am fully agree on this .

The findings of the research has the potential to contribute valuable information to global efforts in combating tuberculosis and drug resistance. Sharing insights on the genomic diversity and transmission patterns of M. tuberculosis strains in Mexico can aid in a better understanding of the global dissemination of drug resistance.

Although the study focused on genomic analysis, it would be beneficial to include experimental data to validate drug resistance mechanisms identified through genomics. This would strengthen the overall conclusions and interpretations of the study. Furthermore, geospatial data is lacking. Addition of this data provides regional variations in drug-resistant tuberculosis distribution and identify specific hotspots of DR TB, which will in fact guide targeted interventions.

Please include details clinical information (e.g., treatment outcomes, patient demographics) with genomic data can provide a more comprehensive view of the epidemiological dynamics of drug-resistant tuberculosis in Mexico. This would offer valuable insights into risk factors and patient-specific management strategies.

It is suggested for sharing the raw genomic data through public repositories .

The study on genomic epidemiology analysis of drug-resistant Mycobacterium tuberculosis distributed in Mexico represents a significant contribution to the field of tuberculosis research. The authors have conducted a thorough investigation, and their findings have important implications for public health. Addressing the suggestions for improvement would strengthen the study's overall impact and contribute to its broader relevance and applicability in combating drug-resistant tuberculosis both in Mexico and globally. Therefore, I would like to recommend this study for publication after reviewing the reviewer suggestion.

Thank you

Thanks for the kind reviewer comments. We would like to say that we have included experimental data of drug resistance determined in the laboratory for each strain. Such data can be found as supplementary information. Unfortunately, we are not able to include detailed geographical information for each sample as all the isolations were obtained at the corresponding health center.

---

## [Editor Report · Decision Letter 1]

3 Oct 2023

Genomic epidemiology analysis of drug-resistant Mycobacterium tuberculosis distributed in Mexico

PONE-D-23-08010R1

Dear Dr. Licona Cassani,

We’re pleased to inform you that your manuscript has been judged scientifically suitable for publication and will be formally accepted for publication once it meets all outstanding technical requirements.

Kind regards,

Dwij Raj Bhatta, PhD

Academic Editor

PLOS ONE

Additional Editor Comments (optional):

All querys and concern from editorial office has been adressed and authors have revised manuscript as suggested by reviewers in resubmitted revision 1 manuscript! Therefore, manuscript revision 1 be accepted for publication
---

## [Editor Report · Acceptance letter]

5 Oct 2023

PONE-D-23-08010R1 

Genomic epidemiology analysis of drug-resistant *Mycobacterium tuberculosis* distributed in Mexico 

Dear Dr. Licona-Cassani:

I'm pleased to inform you that your manuscript has been deemed suitable for publication in PLOS ONE. Congratulations! Your manuscript is now with our production department. 

Kind regards, 

on behalf of

Professor Dwij Raj Bhatta 

Academic Editor

PLOS ONE